# DNA Binding and Cleavage, Stopped-Flow Kinetic, Mechanistic, and Molecular Docking Studies of Cationic Ruthenium(II) Nitrosyl Complexes Containing “NS_4_” Core

**DOI:** 10.3390/molecules28073028

**Published:** 2023-03-28

**Authors:** Hadeer A. Shereef, Yasmine S. Moemen, Fawzia I. Elshami, Ahmed M. El-Nahas, Shaban Y. Shaban, Rudi van Eldik

**Affiliations:** 1Chemistry Department, Faculty of Science, Menoufia University, Shebin El-Kom 32512, Egypt; 2Clinical Pathology Department, University Hospital, Menoufia University, Shebin El-Kom 32512, Egypt; 3Clinical Pathology Department, National Liver Institute, Menoufia University, Shebin El-Kom 32512, Egypt; 4Chemistry Department, Faculty of Science, Kafrelsheikh University, Kafrelsheikh 33516, Egypt; 5Department of Chemistry and Pharmacy, University of Erlangen-Nuremberg, 91058 Erlangen, Germany; 6Faculty of Chemistry, Nicolaus Copernicus University in Torun, 87-100 Torun, Poland

**Keywords:** DNA, ruthenium nitrosyl complexes, reaction mechanism, binding interaction, molecular docking, kinetic investigation

## Abstract

**Highlights:**

Theoretical studies were performed on [RuNOTSP]^+^, TSPH_2_, and its anion TSP^2−^ using the DFT/B3LYP method.Cationic complex [RuNOTSP]^+^ and TSPH_2_ were investigated mechanistically for ctDNA interaction.Spontaneous ctDNA binding via a static mechanism with two steps was reported.Detailed kinetic data are reported and relative reactivity is [RuNOTSP]^+^/TSPH_2_ = 3/1.The ruthenium effect on affinity and mechanism is reported.The ruthenium center improves the reaction rate through coordination affinity, but does not change its mechanism.Molecular docking was used to predict the binding between [RuNOTSP]^+^ and TSPH_2_ and the receptors.DNA cleavage studies are correlated with kinetic data.

**Abstract:**

This work aimed to evaluate in vitro DNA binding mechanistically of cationic nitrosyl ruthenium complex [RuNOTSP]^+^ and its ligand (TSPH_2_) in detail, correlate the findings with cleavage activity, and draw conclusions about the impact of the metal center. Theoretical studies were performed for [RuNOTSP]^+^, TSPH_2_, and its anion TSP^−2^ using DFT/B3LYP theory to calculate optimized energy, binding energy, and chemical reactivity. Since nearly all medications function by attaching to a particular protein or DNA, the in vitro calf thymus DNA (ctDNA) binding studies of [RuNOTSP]^+^ and TSPH_2_ with ctDNA were examined mechanistically using a variety of biophysical techniques. Fluorescence experiments showed that both compounds effectively bind to ctDNA through intercalative/electrostatic interactions via the DNA helix’s phosphate backbone. The intrinsic binding constants (K_b_), (2.4 ± 0.2) × 10^5^ M^−1^ ([RuNOTSP]^+^) and (1.9 ± 0.3) × 10^5^ M^−1^ (TSPH_2_), as well as the enhancement dynamic constants (K_D_), (3.3 ± 0.3) × 10^4^ M^−1^ ([RuNOTSP]^+^) and (2.6 ± 0.2) × 10^4^ M^−1^ (TSPH_2_), reveal that [RuNOTSP]^+^ has a greater binding propensity for DNA compared to TSPH_2_. Stopped-flow investigations showed that both [RuNOTSP]^+^ and TSPH_2_ bind through two reversible steps: a fast second-order binding, followed by a slow first-order isomerization reaction via a static quenching mechanism. For the first and second steps of [RuNOTSP]^+^ and TSPH_2_, the detailed binding parameters were established. The total binding constants for [RuNOTSP]^+^ (K_a_ = 43.7 M^−1^, K_d_ = 2.3 × 10^−2^ M^−1^, ΔG^0^ = −36.6 kJ mol^−1^) and TSPH_2_ (K_a_ = 15.1 M^−1^, K_d_ = 66 × 10^−2^ M, ΔG^0^ = −19 kJ mol^−1^) revealed that the relative reactivity is approximately ([RuNOTSP]^+^)/(TSPH_2_) = 3/1. The significantly negative ΔG^0^ values are consistent with a spontaneous binding reaction to both [RuNOTSP]^+^ and TSPH_2_, with the former being very favorable. The findings showed that the Ru(II) center had an effect on the reaction rate but not on the mechanism and that the cationic [RuNOTSP]^+^ was a more highly effective DNA binder than the ligand TSPH_2_ via strong electrostatic interaction with the phosphate end of DNA. Because of its higher DNA binding affinity, cationic [RuNOTSP]^+^ demonstrated higher cleavage efficiency towards the minor groove of pBR322 DNA via the hydrolytic pathway than TSPH_2_, revealing the synergy effect of TSPH_2_ in the form of the complex. Furthermore, the mode of interaction of both compounds with ctDNA has also been supported by molecular docking.

## 1. Introduction

One of the most important factors in the advancement of medicinal inorganic chemistry is still the widespread use of cisplatin and other platinum-based metallodrugs as chemotherapeutic agents against various cancers, including those of the ovary, bladder, and testicles [1,2,3,4]. In the search for coordination compounds that are toxic-free compared to cisplatin and effective against tumors, ruthenium compounds show the most promise due to their biological properties, which differ greatly from those of traditional platinum compounds in terms of their mechanism of action, toxicity, and biodistribution, and some ruthenium compounds have been shown to be quite selective for cancer cells [5,6,7,8]. It is widely known that ruthenium complexes have a strong affinity for cancer tissues and a reduced general toxicity when they bind to biological molecules such as DNA, making them potential candidates for anticancer drugs [9,10,11,12,13,14,15,16]. This is thought to be caused by ruthenium’s capabilities to mimic iron when binding to biomolecules. Ruthenium-based medications may reach cancer cells more effectively because cancer cells overexpress transferrin receptors to meet their increased need for iron [17].

Few ruthenium complexes have so far been demonstrated to exhibit potential anticancer activities and have been registered in clinical trials [18,19]. These complexes have some advantages over cisplatin, such as their efficacy against cancer cell lines that are resistant to cisplatin and their higher selectivity for cancer cells compared to normal cells, which can lessen side effects [20,21,22,23]. Additionally, ruthenium complex structures may have stronger anticancer activity if biomolecules such as NO are added to them. It has been demonstrated that NO functions as a mediator in one tumor-induced angiogenic process, which is a critical step in the formation of metastasis [24,25,26]. The significance of ruthenium nitrosyl complexes in terms of biology, environment, and novel reactivity has been thoroughly investigated over time [27,28,29,30]. When cationic nitrosyl ruthenium complexes with multi-dentate sulfur ligands are prepared, a variety of complexes with different structures, electronic characteristics, and reactivity trails are created [31,32,33,34]. One of these cationic nitrosyl ruthenium complexes, [RuNOTSP]^+^ {TSP^2−^ = 4-(diethylamino)-2,6-bis[(2-mercaptophenyl)thiomethyl]-pyridine(2-)}, exhibits high activity against cancer cells and is resistant to air oxidation [31].

Because DNA is the primary target of most antitumor agents, the interaction of well-tailored metal complexes with DNA determines the potential of these complexes to act as potent chemotherapeutic agents. DNA is a polyelectrolyte with a high negative charge at physiological pH, and the structure of DNA is governed by electrostatic and hydrophobic interactions between the various residues in the polynucleotide chain [35]. Cationic agents are expected to bind more strongly to DNA, a negatively charged molecule, through a combination of electrostatic attraction, groove binding, and intercalation. Because cationic ruthenium complexes, among other cationic agents, have unique properties such as binding to DNA or protein [36], good luminescent behavior, and singlet oxygen-generating abilities [35], the emphasis has shifted to cationic complexes as complexing agents with DNA. In an effort to comprehend how biomolecules bind in vitro and determine whether the metal center is necessary for binding, the BSA binding characteristics of [RuNOTSP]^+^ and its ligand TSPH_2_ have recently been reported [37] (Figure 1). In the current study, we used cationic [RuNOTSP]^+^ and TSPH_2_ for DNA binding and DNA cleavage behavior to understand the action mechanism of drugs to DNA, which improves the design of drugs that target cellular DNA, in order to learn more about the anticancer activity of ruthenium complexes. This was thoroughly studied using fluorescence spectroscopy, electronic absorption spectroscopy, and gel electrophoresis to show how [RuNOTSP]^+^ and TSPH_2_ interact with DNA as drugs. Molecular docking was used to theoretically investigate the binding mode of [RuNOTSP]^+^ and its ligand TSPH_2_ for DNA binding sites. We also performed a kinetic analysis of the interaction between ctDNA-[RuNOTSP]^+^ and ctDNA-TSPH_2_. In this respect, gene transport is also greatly influenced by the kinetics of DNA condensation. As it is expected that the drug interaction will happen so quickly, a fast kinetic method is needed to find transient intermediates in the interaction pathway. We employed the stopped-flow technique in earlier studies [37,38,39,40,41], in which sample and reagent solutions were quickly mixed, and measurements were taken almost immediately after mixing. Using this technique, it was found that other factors, such as drug affinity and the kinetic stability of the DNA/protein–drug complex, are also essential for biological activity when examining how various drugs interacted with DNA and proteins. In this study, we evaluated the compound–DNA affinity using the stopped-flow method. We also examined the kinetic stability and, finally, proposed an interaction mechanism. The geometrically optimized parameters (bond lengths, bond angles, and dihedral angles) were computed using DFT/B3LYP theory to conduct computational studies on the cationic component [RuNOTSP]^+^, TSPH_2_, and its anion TSP^2−^.

## 2. Results and Discussion

Ruthenium nitrosyl complex [RuNOTSP]Br was prepared in our lab according to the methods described [33]. Starting from 4-diethylamino-2,6-bis(hydroxymethyl)pyridine, the final product 4-diethylamino-2,6-bis(bromomethyl)pyridine was obtained. The latter was added to a solution of NBu_4_[Ru(NO)(S_2_C_6_H_4_)_2_] in boiling THF, giving a brown suspension, from which a brown solid of [RuNOTSP]Br was isolated by filtration. It could be isolated in an analytically pure form, proved soluble in MeOH, CH_2_Cl_2_, and DMF, and its elemental analyses and spectroscopic data (IR, ^1^H, ^13^C NMR, mass spectra) were compatible with those reported. [RuNOTSP]Br is easily hydrolyzed by concentrated hydrochloric acid in CH_2_Cl_2_ at room temperature, yielding the ligand as the pyridinium salt TSPH_2_ HCl (Appendix A). The (NO) wave number of complex [RuNOTSP]Br is 1858 cm^−1^ in KBr, 1870 cm^−1^ in THF, and 1880 cm^−1^ in MeOH. UV–vis spectra in methanol revealed that [RuNOTSP]^+^ is stable over time, with no discernible changes. This indicates that under these conditions, reducing [RuNOTSP]^+^ to [RuNOTSP] is difficult. As a result, under these conditions, nitrosyl ligand (NO^+^) was coordinated to the ruthenium center. Cyclic voltammetry could also help with the difficult reduction of [RuNOTSP]^+^ to [RuNOTSP] in methanol. Within the solvent limits, the cyclic voltammogram of [RuNOTSP]^+^ in MeOH shows two successive reduction processes. The observed redox processes at E_½_ = ca. −467 and −1524 mV vs. NHE (Figure 1) were the most likely, as seen in the two stepwise one-electron reductions designated in Equation (1). The large potential values of [RuNOTSP]^+^ compared to the related complexes (−467, −1524 mV vs. −275, −1200 mV) imply that the Et_2_N substituent makes the [RuNOTSP]^+^ fragment more difficult for reduction. This result is consistent with the differences in (NO) frequencies (KBr) of [RuNOTSP]^+^ (1858 cm^−1^) in comparison to the previously reported complex (1892 cm^−1^) [42].
(1)$${\left\{\mathrm{RuNOTSP}\right\}}^{6}\begin{array}{c} +{\;e}^{-}\\ ⇌\\ -{\;e}^{-} \end{array}{\left\{\mathrm{RuNOTSP}\right\}}^{7}\overset{+{\;e}^{-}}{\to }{\left\{\mathrm{RuNOTSP}\right\}}^{8}$$

Three oxidation processes were observed in the anodic region at E_½_ = ca. +560, +915, and +1207 mV, which could be tentatively assigned to the three stepwise one-electron oxidations designated in Equation (2).
(2)$${\left[\mathrm{RuNOTSP}\right]}^{+}\begin{array}{c} +{\;e}^{-}\\ ⇌\\ -{\;e}^{-} \end{array}{\left[\mathrm{RuNOTSP}\right]}^{2+}\overset{-\;\mathrm{NO}}{\to }{\left[\mathrm{RuTSP}\right]}^{2+}\begin{array}{c} +{\;e}^{-}\\ ⇌\\ -{\;e}^{-} \end{array}{\left[\mathrm{RuTSP}\right]}^{3+}$$

### 2.1. Computational Findings

Figure 2 depicts the optimized geometries of the investigated TSPH_2_, TSP^2−^ anion, and [RuNOTSP]^+^. These calculations reveal that TSPH_2_ has a square-pyramidal structure and pseudo-octahedral overall geometry around the metal center. The two thioether and two thiolate donor atoms of the TSPH_2_ ligand occupy the corresponding trans position and provide the steric rigidity of the py(CH_2_)_2_ backbone, as does the bridging S-donor and TSPH_2_ or the pyridine N-donor of the metal fragment.

To shed light on the electronic structures of TSPH_2_, TSP^2−^ anion, and [RuNOTSP]^+^, the natural population analysis (NPA) was carried out [43,44,45]. Atomic numbering along with the data are listed in the Appendix A. Table 1 shows charges from natural population analysis over some selected atoms. The negative charge on the thiolate sulfur atoms of TSP^2−^ anion (S38, −0.477 and S48, −0.477) is larger than that for the neutral ligand TSPH_2_ (S38, 0.066 and S48, 0.066), whereas both are larger than that of the cationic complex [RuNOTSP]^+^ (S12, 0.121 and S24, 0.1563). This indicates that the two negative charges on the thiolate of TSP^2−^ are shifted towards the ruthenium center after complexation. The same trend is also valid for the thioether sulfur atoms of TSP^2−^ anion (S10, S26), TSPH_2_ (S10, S26), and cationic complex [RuNOTSP]^+^ (S11, S23). So, the charge density on the thiolate and thioether sulfur atoms is delocalized over the ruthenium center and makes the ruthenium center more negative (Ru, −0.674). The charge density on the pyridine nitrogen atom (N26, −0.43862) is more negative compared to that of the nitrogen atom of the nitrosyl group, which indicates that the positive charge on the complex is more concentrated in the nitrosyl group.

#### 2.1.1. Quantum Chemical Parameters

In a complex formation system, the ligand serves as an electron donor (Lewis base), and the metal ion serves as an electron acceptor (Lewis acid). A ligand with the correct softness value usually chelates metal ions successfully [46]. Additionally, the chemical reactivity, stability, and hardness of the compounds are explained by the energy difference between HOMO and LUMO. A hard molecule is one with a large HOMO-LUMO gap, whereas a soft molecule is one with a small HOMO-LUMO gap. While softness gauges chemical reactivity, hardness gauges molecule stability. From the data of HOMO and LUMO energies, the energy gap (ΔE), absolute electronegativity (χ), absolute hardness (η), electronic chemical potentials (Pi), absolute softness (σ), global softness (S), and global electrophilicity (ω) are listed in Table 2. The comparative investigation clarifies the following points in association with TSPH_2_: (i) the soft character of TSPH_2_ shows its flexible reactivity toward the metal ions; (ii) χ is positive whereas Pi is negative values indicated that the molecule is able to capture electrons from its environment and its energy must decrease when accepting the electron charge. The quantum chemical parameters for [RuNOTSP]^+^ elicit the following points: (i) the energy gaps of the cation [RuNOTSP]^+^ in comparison with TSPH_2_ reflect the high softness and biological activity of metal complexes than TSPH_2_; (ii) E_HOMO_ increase and E_LUMO_ more than TSPH_2_, which might be related to the strength of metallic bonds.

#### 2.1.2. Frontier Molecular Orbitals and Chemical Reactivity

Frontier molecular orbitals (FMOs) are used to understand several types of reactions and explain the interaction between the ligand and the metal ion, besides predicting the most reactive position in the molecules and their properties. The HOMO level is the highest energy orbital containing electrons that act as an electron donor, while the LUMO is the lowest energy orbital that acts as an electron acceptor. The calculated HOMO of TSPH_2_ is located on C_3_, C_4_, C_5_, C_6_, C_8_, C_9_, C_10_, C_11_, N_1_, N_2_, S_1,_ and S_2_, whereas the HOMO of the TSP^2−^ anion spreads over two benzene rings, N_1_, S_1_, S_2_, S_3,_ and S_4_. The LUMO of TSPH_2_ and TSP^2−^ anion is distributed over the whole skeleton, as shown in Figure 3. These orbitals over the relevant centers indicate the sites that can act as an electron donor in TSPH_2_ when coming near the metal ion and those that can accept the electron via back donation from the metal ion to form stable complexes. Deprotonation of the thiol groups further adds new centers that share HOMO orbitals that become ready for binding with interacting metal ions. The HOMO level of [RuNOTSP]^+^ spreads over two benzene rings, Ru, N_3_, O_1_, S_3,_ and S_4_, whereas LUMO is located on Ru, N_3_, O_1_, S_1,_ and S_2_ (Figure 3).

Molecular electrostatic potential (MEP) is a very suitable tool for highlighting reactive sites towards electrophilic and nucleophilic attack and also determines the relative polarity of molecules. The MEPs of TSPH_2_, and [RuNOTSP]^+^ are presented in Figure 4. The red and yellow sites (negative regions) of MEP are related to electrophilic reactivity, while the blue and green sites (positive regions) are related to nucleophilic reactivity. Figure 4 reveals that the negative MEP regions of TSPH_2_ are concentrated on the pyridine nitrogen atom and sulfur atoms, thus confirming their electron donor ability.

### 2.2. DNA Interaction Studies

#### 2.2.1. Fluorescence Spectroscopy

The most accurate and sensitive method for characterizing how drugs interact with CT-DNA is fluorescence emission spectra [47,48]. Both [RuNOTSP]^+^ and TSPH_2_ exhibit a noticeable increase in emission when CT-DNA is added because they are both luminescent in the absence of DNA. Figure 5 demonstrates that increasing the concentration of DNA at 25 °C and pH = 7.2 resulted in a regular increase in the fluorescence intensity of both compounds without shifting the fluorescence emission maximum. With increasing DNA concentration, [RuNOTSP]^+^ and TSPH_2_ solution’s maximum fluorescence intensity increases steadily at 303 nm, with no discernible change in the position or shape of the emission bands. The increase in emission intensity is most likely caused by the modification of the compound’s environment and the degree to which the compound has been inserted into the hydrophobic environment inside the DNA helix. Fewer solvent molecules can reach the binding site due to the hydrophobic environment inside the DNA helix, preventing the quenching effect. The DNA helix effectively protected both compounds, as evidenced by the fact that relaxation vibrations decreased and emission intensity increased. The increase in emission intensities reveals their binding to DNA into its hydrophobic pocket along the major and minor grooves [49,50]. These fluorescence improvements demonstrate that [RuNOTSP]^+^ and TSPH_2_ both interacted with DNA to increase the quantum efficiency of the complex. Similar to the quenching process, Equation (3) can be used to determine the enhancement constant [51,52]:
(3)$$I_{0}/I=1\;–\;K_{E}[E]$$

Equation (3) can be written in the case that a dynamic process contributes to the enhancing mechanism, as follows (Equation (4)):

(4)$$I_{0}/I=1\;–\;K_{D}\left[E\right]=1\;–\;k_{B}\;\tau_{0}\left[E\right]$$
where k_D_ is the dynamic enhancement constant (similar to a dynamic quenching constant), k_B_ is the bimolecular enhancement constant (similar to a bimolecular quenching constant), and τ_0_ is the lifetime of the fluorophore in the absence of the enhancer, which equals about 10^−9^ s [53], and [E] is the concentration of enhancer. The plot of I_0_/I vs. [E], which gives the K_D_ values by slope, and the values of k_B_ are calculated by using Equation (4), and these results are given in Figure 6a and summarized in Table 3. The plots exhibited a linear relationship, which indicated only one type of enhancing (static or dynamic enhancing). The dynamic constant (K_D_) of [RuNOTSP]^+^ (3.3 ± 0.3) × 10^4^ M^−1^ is higher than that of its ligand TSPH_2_ (2.6 ± 0.2) × 10^4^ M^−1^. The enhancement rate constant of the biomolecule (k_B_) of [RuNOTSP]^+^ and TSPH_2_ is calculated as (3.3 ± 0.3) × 10^12^ and (2.6 ± 0.2) × 10^12^ L·mol^−1^·s^−1^ at 298 K, respectively. When the equivalence of the bimolecular quenching and enhancement constants are considered, the latter is found to be greater than the maximum possible value (1 × 10^10^ L·mol^−1^·s^−1^) in aqueous medium. Thus, the fluorescence enhancement is not initiated by a dynamic process; rather, a static process involving complex formation in the ground state is proposed, and the fluorescence enhancement is controlled by a static process [50,51]. The calculated association constants for the ctDNA-TSPH_2_ and ctDNA-[RuNOTSP]^+^ adducts indicate high-affinity TSPH_2_- and [RuNOTSP]^+^-polynucleotide binding. 

#### 2.2.2. Equilibrium Binding Titration

The following Equation (5) was used to determine the binding constant (K_f_) and the number of [RuNOTSP]^+^ and TSPH_2_ molecules bound per polynucleotides (n) for the complex formation between both compounds and DNA [54]:(5)$$\log\left(\frac{I_{0}-I}{I}\right)={\mathrm{logK}}_{f}+\mathrm{nlog}\left[\mathrm{DNA}\right]$$
where I_0_ and I are the fluorescence intensity of the fluorophore in the absence and presence of different concentrations of DNA, respectively. The linear equations of log (I – I_0_)/I vs. log [DNA] are shown in Figure 6b. The K_f_ values demonstrate the complex’s remarkably high affinity for DNA, and the n values from the slope of the straight line are 0.9 and 0.7 for ctDNA-[RuNOTSP]^+^ and ctDNA-TSPH_2_ adducts, respectively.

#### 2.2.3. UV–Vis Spectroscopy

One of the most effective methods for studying DNA binding is electronic absorption spectroscopy [55]. The π-π* transitions of DNA bases are what cause the DNA band to appear at 260 nm. Changes in the stacking pattern, dissolution of the hydrogen bonds between complementary strands, covalent binding of DNA bases, and intercalative mode, which involves a strong stacking interaction between aromatic rings of molecules and the base pairs of DNA, are the causes of the hypochromism, red, and/or blue shifts of this band [56,57]. In Tris buffer, the electronic spectrum of TSPH_2_ is characterized by an intense ligand-centered transition in the UV region at 276 nm, which is blue shifted in [RuNOTSP]^+^ and appears at 272 nm. This ultraviolet band can be attributed to π-π* transition. Furthermore, a metal-to-ligand charge transfer (MLCT) in the visible region at 310 nm for [RuNOTSP]^+^ was observed, which was attributed to the overlap of Ru(d_π_)-TSP (π*). [RuNOTSP]^+^ exhibited a visible band at 514 nm that can be attributed to the d-d transition (Figure 7).

Figure 8 shows the absorption spectra of [RuNOTSP]^+^ and TSPH_2_ in the absence and presence of increasing amounts of CT-DNA. The UV–vis absorption of [RuNOTSP]^+^ and TSPH_2_ is significantly perturbed by the addition of increasing amounts of CT-DNA. In detail, the UV absorption bands of [RuNOTSP]^+^ (272 nm) and TSPH_2_ (276 nm) show a red shift of about 3 nm and hypochromism (Figure 8). For [RuNOTSP]^+^, a clear isosbestic point was observed at 329 nm, whereas for TSPH_2_, the isobestic point comes at 352 nm (Figure 8). Similar spectral properties, such as bathochromic shift and hypochromism in the presence of CT-DNA, have been observed for ruthenium (II) complexes and have been linked to a mode of binding involving an interaction of aromatic chromophores with the base pairs of ctDNA that stacks [58]. Using Equation (6) [59] and Figure 9, it was discovered that [RuNOTSP]^+^ and TSPH_2_ have intrinsic binding constants, K_b_, of (2.4 ± 0.2) × 10^5^ M^−1^ and (1.9 ± 0.3) × 10^5^ M^−1^, respectively. These values are higher than those reported by Barton and colleagues [60] for related ruthenium complexes (1.15 × 10^4^ M^−1^). Furthermore, the K_b_ values are very close to the binding constants of the delta and lambda isomers of [Ru(*o*-phen)_3_]^2+^, which were assigned as non-intercalators [61]. This result indicated that [RuNOTSP]^+^ binds to DNA stronger than TSPH_2_.
(6)$$\frac{1}{A_{\mathrm{obs}}-A_{0}}=\frac{1}{A_{c}-{\;A}_{0}}+\frac{1}{K_{b}\left(A_{c}-{\;A}_{0}\right)\left[\mathrm{DNA}\right]}\;$$
where A_obs_, A_0_, and A_c_ represent the observed absorbance during the interaction, the absorbance of DNA only, and the absorbance of DNA with the compound, respectively.

#### 2.2.4. Stopped-Flow Spectroscopic Studies and Kinetic Investigation

Any potential polycation gene delivery agent must meet two crucial requirements: (a) the capacity to completely neutralize the charge of the native DNA, resulting in a more compact state for the polycomplexes formed by DNA and polycations; and (b) the ability to dissociate the DNA-polycation complexes in the target cell’s cytoplasm [62,63,64]. The kinetic parameters of the DNA polycomplex formation process may hold some crucial information in this regard. To keep track of the process’ kinetic parameters, we used the stopped-flow method. Figure 10 uses typical kinetic traces to show the formation of the ctDNA-[RuNOTSP]^+^ and ctDNA-TSPH_2_ complexes after [RuNOTSP]^+^ and TSPH_2_ have bound to the ctDNA in a 10 mM phosphate buffer solution. By fitting the kinetic curves to an exponential sum, we were able to analyze the multi-exponential interaction (Equation (7)).
(7)$$A_{t}=a_{1}e^{-k_{\mathrm{obs}1}t_{1}}+a_{2}e^{-k_{\mathrm{obs}2}t_{2}}+A_{0}$$
where A_t_ is the absorbance intensity at time t. Data analysis was performed using Origin 8.0 software. The number of exponentials was increased until there was no discernible systematic deviation of the residual. Figure 10 shows experimental plots of the absorption intensity of CTDNA-[RuNOTSP]^+^ and ctDNA-TSPH_2_ at 260 nm as a function of time at 25 °C for each of the two compounds. Table 4 shows the observed rate constants k_obs_ as a function of [RuNOTSP]^+^ and TSPH_2_ concentrations at constant DNA concentration and temperature. The average of three separate experiments serves as the rate constant. The values of these rates were at least an order of magnitude apart for all of the [RuNOTSP]^+^ and TSPH_2_ under study. ctDNA-[RuNOTSP]^+^ and ctDNA-TSPH_2_ complexes were formed via a bimolecular mechanistic pathway. From the exponential plots that followed the order k_1_ ˃˃ k_2_, we were able to determine two relative rate constants (k_1_ and k_2_) for the binding process. All kinetic curves were able to fit into the bi-exponential function (Equation (7)), indicating the formation of complexes involved in a two-step process. The initial step is the fast step (rate constant k_obs1_), in which [RuNOTSP]^+^ and TSPH_2_ bind to ctDNA in a reversible reaction to form the binary intermediate via electrostatic interactions. The initial reaction step is thought to be the complexation and production of binary complexes. The second step (rate constant k_obs2_) of [RuNOTSP]^+^ and TSPH_2_ is slower than the first, involving compaction of the ctDNA and simultaneous internal rearrangement of the CTDNA-[RuNOTSP]^+^ and CTDNA-TSPH_2_ complexes [65,66,67].

Figure 11 depicts plotting of k_obs1_ vs. the concentration of [RuNOTSP]^+^ and TSPH_2_ and for the first step gave a linear relationship with a slope k_on_ [M^−1^ s^−1^] and an intercept k_off_ [s^−1^], and the observed rate constant can be expressed by Equation (8).
(8)$$k_{\mathrm{obs}1}=k_{\mathrm{off}}+k_{\mathrm{on}}\left[\mathrm{RuNOTSP}\right]$$

Both k_on_ and k_off_ were used to obtain the equilibrium association constants (K_a_ [M^−1^] = (k_on_/k_off_) and the equilibrium dissociation constants (K_d_ [M] = k_off_/k_on_). As shown in Figure 11, the k_obs2_ values for the second step were also found to increase with increasing [RuNOTSP]^+^ and TSPH_2_ concentration, although the increase was less compared to that of k_obs1_. This is expected as the second step involves the compaction of the large CT-DNA molecule. [RuNOTSP]^+^ binds to ctDNA with a second-order association constant of k_1_ = 1.0 ± 0.2 M^−1^ s^−1^ and dissociates from the binary complex with a first-order dissociation constant of k_−1_ = (4.1 ± 0.1) × 10^−2^ s^−1^, whereas TSPH_2_ binds to CT-DNA with (k_1_ = 0.7± 0.1 M^−1^ s^−1^) and dissociates from the binary complex with (k_−1_ = (5.5 ± 0.0) × 10^−2^ s^−1^). This means that the DNA binding affinity of [RuNOTSP]^+^, K_a1_, k_1_/k_−1_ is much higher (45.6 M^−1^) than that of TSPH_2_ (12.7 M^−1^) and that the equilibrium dissociation constant of [RuNOTSP]^+^, K_d1_, k_−1_/k_1_ is 2.2 × 10^−2^ M, which is much lower than TSPH_2_ (7.9 × 10^−2^ M). The presence of cationic [RuNOTSP]^+^ in the negatively charged ctDNA solution facilitated a faster interaction, leading to a faster binding between the [RuNOTSP]^+^ and ctDNA compared to the neutral TSPH_2_, and as a result, the formed ctDNA-[RuNOTSP]^+^ is much more stable than ctDNA-TSPH_2_. The higher negative G_bind_ value of [RuNOTSP]^+^ (−17.1 kJ mol^−1^) compared to TSPH_2_ (−6.7 kJ mol^−1^) supports the higher stability of ctDNA-[RuNOTSP]^+^ compared to ctDNA-TSPH_2_.

The data from the second reaction step, which included internal DNA rearrangement as well as electrostatic interaction and isomerization reaction, followed the same pattern as the first; a reversible reaction was observed for both [RuNOTSP]^+^ and TSPH_2_. The DNA affinity for [RuNOTSP]^+^ (K_a2_ = k_2_/k_−2_) is 40.0 M^−1^, which is slightly lower than TSPH_2_ (59 M^−1^), and the equilibrium dissociation constant for RuNOTSP (K_d2_ = k_−2_/k_2_) is 2.5 × 10^−2^ M, which is slightly higher than TSPH_2_ (1.7 × 10^−2^ M). G_bind_ values for this phase are −19.5 kJ mol^−1^ for [RuNOTSP]^+^ and −13.2 kJ mol^−1^ for TSPH_2_, indicating that this step is spontaneous for both compounds. The increment in this step was less than that in the first stage. Actually, the second step closely resembled the DNA condensation process, where electrostatic interaction was no longer the sole determining factor. Internal rearrangements within the DNA secondary structure occurred during this step, resulting in DNA chain compaction. This was, indeed, a slower process than the first. A comparison of the overall coordination affinity K_a1_ of [RuNOTSP]^+^ (43.7 M^−1^) and TSPH_2_ (15.1 M^−1^) and equilibrium dissociation constants [RuNOTSP]^+^ (K_d1_ 2.3 × 10^−2^ M) and TSP (66 × 10^−2^ M) revealed that [RuNOTSP]^+^ binds to CT-DNA much tighter than TSPH_2_. This suggests that the first interaction step, complexation and the formation of binary complexes, is far more important in determining the rate than the second interaction step, DNA condensation (See Figure 2).

### 2.3. DNA Cleavage Studies

Apoptosis is a distinct morphological and biochemical process that results in cell death and is induced by a number of physiological and pathological factors. Chemotherapeutic agents cause apoptosis, which partially kills cells. The quantification of this process allows for the study of various cellular responses to chemotherapy and possibly clinical sensitivity. The apoptotic process causes cell shrinkage, membrane blabbing, nuclear condensation, and inter-nucleosome DNA fragmentation. There are numerous ways to identify apoptosis along this pathway. The ctDNA cleavage abilities of [RuNOTSP]^+^ and TSPH_2_ were measured using a diphenylamine assay. Generally, single-strand or double-strand breaks can convert the plasmid DNA from supercoil DNA to other forms of DNA [68,69]. The diphenylamine assay results are shown in Figure 12. For control experiments, untreated DNA showed a little cleavage upon irradiation (~4%). However, the presence of [RuNOTSP]^+^ or TSPH_2_ could cleavage DNA effectively. Compared to TSPH_2_, cationic [RuNOTSP]^+^ displayed higher cleavage efficiency against plasmid DNA due to its stronger DNA binding affinity, revealing the synergistic effect of TSPH_2_ in the form of the complex.

### 2.4. Molecular Docking of DNA with [RuNOTSP]^+^ and TSPH_2_

The molecular docking technique introduces a small molecule into the binding site of the DNA target-specific region primarily in a non-covalent mode, which can aid in rational drug design and mechanistic studies. Interaction mode and binding affinity docking studies on TSPH_2_ and [RuNOTSP]^+^ with B-DNA have been carried out to investigate the most likely binding site [70] (PDB ID: 1BNA). For molecular docking, we used the new server PatchDock [71], and the best docking model is shown in Figure 13 and Figure 14. As the DNA fragment (2L8i) contains a triazole template, which has a far better affinity and specificity than unmodified oligonucleotides, and it is also highly resistant to nuclease destruction [72]. Both compounds [RuNOTSP]^+^ and TSPH_2_ are docked to the major groove of the DNA fragment (2L8i), as seen in (Figure 13 and Figure 14), where [RuNOTSP]^+^ binds to DNA through different types of interactions as one sulfur atom forms one hydrogen bond (HB) with strand B (SB), adenine Base Strand (A) (SA) Number 16; also one HB between oxygen and Cytosine (C) base SA No. 9; van der Waals interaction between [RuNOTSP]^+^ and different bases such as C No. 12 SA, Guanine (G) No. 13 SA, C No. 17 SB, G No. 18 SB; π-π stacking interaction between phenyl group and G No. 10 SA; π-sulfur stacking interaction with sulfur atom and Thymine (T) No. 11 SA; pi-anion interaction with G No. 15 SB and steric effect at nitrogen and C No. 14 SB. These different types of interaction for [RuNOTSP]^+^ are stronger and more stable than TSPH_2_, as shown in (Figure 13 and Figure 14). TSPH_2_ binds to DNA by several types of interactions as two sulfur atoms forms: 2 HB with SB G Number 18; HB between sulfur and triazole linker No. 19 [73]; van der Waals interaction between TSPH_2_ and different bases such as C No. 12 SA, T No. 11 SA, and T No. 20 SB; HB with nitrogen and G No. 10 SA; π-sulfur stacking interaction with sulfur atom and carbon of the TSPH_2_ and steric effect at nitrogen and C No. 9 SA. The hydrogen bond acceptor sites of the DNA phosphate group in combination with DNA polymerase can thus be mimicked using the triazole template strand [53] or TSPH_2_, and this will have an effect on DNA bending, rigidity, and recognition. In a previous study [41], the docking results showed that BSA-[RuNOTSP]^+^ with binding affinity (−7.27 kcal/mol) and BSA-TSPH_2_ with binding affinity (−8.05 kcal/mol) are located in subdomain IA and it has recently been reported as a possible binding site for drugs on BSA [74].

## 3. Conclusions

In the current study, the interaction of DNA with [RuNOTSP]^+^ and its ligand TSPH_2_ is studied by fluorescence quenching, UV–vis absorption studies, stopped-flow, and molecular docking. The fluorescence enhancement spectra of both compounds are controlled by a static enhancement mechanism. The intrinsic binding constants (K_b_), (2.4 ± 0.2) × 10^5^ M^−1^[RuNOTSP]^+^ and (1.9 ± 0.3) × 10^5^ M^−1^ (TSPH_2_), as well as the enhancement dynamic constants (K_D_), (3.3 ± 0.3) × 10^4^ M^−1^ and (2.6 ± 0.2) × 10^4^ M^−1^ (TSPH_2_), reveal strong electrostatic binding via the phosphate backbone of the DNA helix. Stopped-flow experiments showed that both compounds bind through two reversible steps: a fast second-order binding, followed by a slow first-order isomerization reaction via a static quenching mechanism. The total binding constants for [RuNOTSP]^+^ (K_a_ = 43.7 M^−1^, K_d_ = 2.3 × 10^−2^ M^−1^, ΔG^0^ = −36.6 kJ mol^−1^) and TSPH_2_ (K_a_ = 15.1 M^−1^, K_d_ = 66 × 10^−2^ M, ΔG^0^ = −19 kJ mol^−1^) revealed that the relative reactivity is approximately ([RuNOTSP]^+^)/(TSPH_2_) = 3/1. The significantly negative ΔG^0^ values are consistent with a spontaneous binding reaction to both [RuNOTSP]^+^ and TSPH_2_ with [RuNOTSP]^+^ is very favorable. Cationic [RuNOTSP]^+^ exhibited preferential efficient cleavage activity towards the minor groove of pBR322 DNA and via the hydrolytic pathway compared to TSPH_2_. The results revealed that the Ru(II) center was found to influence the rate but not the mechanism and showed that the [RuNOTSP]^+^ complex is more prominent in DNA binders than the ligand TSP due to the strong electrostatic interaction of the [RuNOTSP]^+^ cation towards the phosphate end of DNA. The optimized geometric structures of the ligand and its metal complexes are in good agreement with the experimental results. Molecular docking studies show that [RuNOTSP]^+^ and TSPH_2_ interact in a parallel manner with the major groove of the DNA backbone through non-covalent interactions such as hydrogen bonding, van der Waals, and hydrophobic interactions. The obtained data indicate that [RuNOTSP]^+^ has a higher activity than TSPH_2_.

## 4. Experimental Section

### 4.1. Materials

We prepared [RuNOTSP]^+^ and TSPH_2_, and then used UV–vis, IR, electrical, and mass spectroscopy studies to characterize them. These outcomes have been compared with those from our studies that have been published [33]. The stock solutions (1 × 10^−3^ M) of [RuNOTSP]^+^ and TSPH_2_ were prepared using methanol, which was then diluted to create working solutions for various spectroscopic techniques. Sigma (Saint Louis, MO, USA), provided the CT-DNA, which was then used without further purification. The stock solution of CT-DNA was made by dissolving a specific quantity of CT-DNA in Tris-HCl buffer at pH 7.2 and room temperature. The solution was then stored at 4 °C in the dark and used right away. With the aid of UV absorption and a molar extinction coefficient of (*ε*) equal to 6600 M^−1^ cm^−1^ at 260 nm, its concentration was ascertained [75].

### 4.2. Apparatus

A JASCO spectrophotometer (FP-8300) using quartz cuvettes of 1 cm was used to measure the absorption spectra of CT-DNA in the absence and presence of the current compounds with different concentrations of [RuNOTSP]^+^ and TSPH_2_, which were recorded at 270–500 nm. Fluorescence measurements were carried out with a JASCO spectrofluorimeter (FP 6200) by keeping the concentration of RuNOTSP]^+^ and TSPH_2_ constant (10^−^^5^ M) while varying the ctDNA concentration from 0 to 1.5 × 10^−^^5^ M.

### 4.3. Stopped-Flow Kinetic Studies

The kinetic measurements were carried out with the Applied KinetAsyst SF-61DX2 stopped-flow instrument (HI-Tech Scientific, Kolkata, West Bengal, India) and a Peltier thermostat. [RuNOTSP]^+^ and TSPH_2_ solutions were used at various concentrations. In the first step, solutions of DNA, and [RuNOTSP]^+^ or TSPH_2_ were added to each of the two syringes of the kinetic accessory. Then, equal volumes of both solutions were simultaneously injected into the sample chamber, and this process was repeated for each run. The DNA’s absorption spectra were continuously monitored before (*t* = 0 s) and after the injection in the presence of [RuNOTSP]^+^ and TSPH_2_. When the instrument’s dead time was measured for a 1:1 mixture, it was found to be 2 ms. In control experiments, DNA solution was combined with buffer solutions devoid of [RuNOTSP]^+^ or TSPH_2_. Decomposition of the DNA was ruled out because the absorption signal of the DNA did not change over the course of the control experiment. The complex nature of the kinetic trace indicated the presence of multiple steps, so we assumed a superposition of exponential terms to express the process (Equation (7)).

### 4.4. ctDNA Cleavage Activity

Chemosensitivity testing involved the DNA fragmentation assay with diphenylamine [76]. In detail: Harvest monolayer cultures by scraping cells into the medium with a rubber policeman (or, for suspension cultures, directly into centrifuge tubes) and centrifuging (300× *g*) at 4 °C for 10 min to pellet the cells. The cell pellet was dissolved in 0.7 mL of ice-cold lysis buffer and 0.8 mL of 10 mM phosphate-buffered saline (PBS) with pH 7.4. The cell lysate was put into microfuge tubes and allowed to sit there for 15 min on ice. After 15 min at 4 °C, the 13,000× *g* lysate was centrifuged to separate high-molecular-weight DNA from DNA that had been broken up. The supernatant was transferred to a 5 mL glass tube (about 1.5 mL containing fragmented DNA). The intacted DNA pellet was suspended in 1.5 mL of Tris-acetate-EDTA (TE) buffer before being transferred to another 5 mL glass tube. Each tube received 1.5 mL of 10% trichloroacetic acid (TCA), which was incubated for 10 min at room temperature and then centrifuged (500× *g*) at 4 °C for 15 min and the supernatant discarded. The 10% TCA precipitates were re-suspended in 0.7 mL of 5% TCA, boiled for 15 min at 100 °C, cooled to room temperature, and centrifuged (300× *g*) for 15 min at 4 °C. Without disturbing the precipitate, 0.5 mL of the supernatant was transferred to a fresh glass tube. After adding 1 mL of the diphenylamine reagent, the absorbance at 600 nm was measured after an overnight incubation at 30 °C. Percentage of DNA fragmentation was calculated as follows:(9)$$\mathrm{DNA}\;\mathrm{fragmentation}\;\left(\%\right)=\frac{{\mathrm{OD}}_{600}\;\mathrm{of}\;\mathrm{the}\;\mathrm{supernatant}}{{\mathrm{OD}}_{600}\;\mathrm{of}\;\mathrm{the}\;\mathrm{supernatant}+{\;\mathrm{OD}}_{600}\;\mathrm{of}\;\mathrm{the}\;\mathrm{pellet}}\times 100\;$$

### 4.5. Computational Studies

The Gaussian 09 W package [77] was used to calculate quantum parameters using density functional theory (DFT) [78,79]. The singlet state of the complexes’ molecular geometry was optimized at the B3LYP level of theory [80,81] using the basis set 6–31 g(d) for the ligand TSPH_2_ and LAN2DZ [49] for the metal in [RuNOTSP]^+^. In the gas phase state, all calculations were optimized, and the optimized coordinates are listed in Appendix A. Frequency calculations for each optimized geometry were done at the same theoretical level to verify if the compound is a minimum or in the transition state. Natural population analysis (NPA) [43,44,45] was performed at B3LYP/6–311 G + (d.p.) using the NBO program version 3.1 [82] to calculate the atomic charge and their impact on the structure and stability of our systems, and the full data are listed in Appendix A. The ChemCraft program was used to display the distribution of the highest occupied molecular orbital (HOMO), lowest unoccupied molecular orbital (LUMO), and frontier molecular orbital (FMO), which were calculated for TSPH_2_ and its complex [83]. The Gaussian view package was used to analyze the molecular electrostatic potential (MEP) [84].

### 4.6. Molecular Docking

Density functional theory (DFT) B3LYP was used to optimize the structures of TSPH_2_ and [RuNOTSP]^+^. The B-DNA crystal data (PDB 1D:1BNA) were obtained from the Protein Data Bank. A geometry-based molecular docking method PatchDock was used [71]. Its goal is to perform molecular docking procedures, which result in favorable complementarity of molecular shapes. Such modifications lead to interface region interactions and minor steric conflicts when they are applied. A wide interface is produced as a result of the multiple complementary local features of the linked molecules that match each other. The Connolly dot surface representation of the molecules [85,86] is divided into concave, convex, and flat patches by the PatchDock method. Following that, complementary patches were matched to create candidate complexes. A scoring formula that takes into account both geometric fit and atomic desolvation energy was used to further assess each feasible change [87]. Finally, using root mean square deviation (RMSD), the candidate complexes were clustered to remove redundant structures. Due to the large part of PatchDock’s quick transformational search, which relies on local feature matching rather than exhaustively searching the six-dimensional transformation space, it is highly efficient. By utilizing powerful data structures and spatial pattern recognition algorithms created in the field of computer vision, such as geometric hashing and posture clustering, and as described in the literature [88], it reduces computational processing time even further.

## Data Availability

Not applicable.

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
