# Peer review of "DNA Binding and Cleavage, Stopped-Flow Kinetic, Mechanistic, and Molecular Docking Studies of Cationic Ruthenium(II) Nitrosyl Complexes Containing “NS4” Core"

_molecules, 2023, doi:10.3390/molecules28073028_

Round 1
Reviewer 1 Report
The manuscript content was done reasonably.
However, I have two questions to fulfill the acceptance of the manuscript.
1. How can the authors parameter the ruthenium atom in molecular docking?
Does the program contain the force field parameter of ruthernium atom and DNA in the docking software? Please clarify this point and state this in the manuscript in the method.
2. Why the study did not need the molecular dynamics simulation of the compound-DNA complex? Could the author conclude or infer that just the molecular docking study was well defined for the molecule interaction of the compound and DNA groove?
Reviewer 2 Report
The authors have reported “DNA Binding and Cleavage, Stopped-Flow Kinetic, Mechanistic, and Molecular Docking Studies of Cationic Ruthenium(II) Nitrosyl Complexes Containing "NS4" core”. In this work, different approaches were used to study the binding affinity of the ligand and the metal complex to DNA. Theoretical studies were performed for the investigated compound. The authors need to correct several points in the manuscript and answer several questions
1. Abstract, the authors amid to answer the question: Is the metal center always important for binding? The question is not formed correctly and must be changed or deleted. The free ligand showed good binding while the coordinated complex enhanced and improved its ability. This is well known in literature and this work also revealed that.
2. The introduction is not clear. It showed another aim that described non-viral gene delivery and gene transfection and the DNA-cationic agent complexes drawbacks. Although ruthenium complexes have good advantages, but they still have high toxicity by cleaving the DNA which is a target to bind, condensing and delivering. Thus, I suggest improving the introduction to be ((study the binding to DNA)).
3. Update the references with recent papers.
4. Some sentences need to be corrected or edited, the abbreviation of BSA, lines 231-233 have another font size, the equations and their numbering.
5. What is the meaning or the purpose of the sentence in line 250?
6. Figure numbering in lines 249 and 300 is wrong and almost all in the paper.
7. Change the optimized structure figures and use Gaussian View to visualize the structures and for the HOMO and LUMO orbitals also with white background.
8. Provide how you calculate the quantum chemical parameters with references.
9. Line 278 ((All quantum parameters are ordered…..)) according what?
10. Line 330 ((Equation (3) can be written…..)) please check the number.
11. Add the pages for references 7, 76.
Round 2
Reviewer 2 Report
Thanks for your great work the paper is ready for publishing
Author Response
Thank you very much for your valuable comments.